# No evidence for parallel evolution of cursorial limb adaptations among Neogene South American native ungulates (SANUs)

**Darin A. Croft** [1]*, **Malena Lorente** [2]

**1** Department of Anatomy, Case Western Reserve University, Cleveland, Ohio, United States of America,
**2** Conicet-División Paleontología Vertebrados, Museo de La Plata (UNLP), La Plata, Argentina

* dcroft@case.edu

**Data Availability Statement:** All relevant data are within the manuscript and its Supporting information files.

**Funding:** The author(s) received no specific funding for this work.

## Abstract

During the Neogene, many North American ungulates evolved longer limbs. Presumably, this allowed them to move more efficiently or quickly in open habitats, which became more common during this interval. Evidence suggests that open habitats appeared even earlier in South America, but no study to date has investigated whether the ungulate-like mammals of South America (South American native ungulates or SANUs) evolved similar limb adaptations. We analyzed limb elongation in the two predominant SANU groups, notoungulates and litopterns, by compiling genus-level occurrences from the late Oligocene to the Pleistocene and calculating metatarsal/femur ratio (Mt:F). None of the groups or subgroups we analyzed show a pronounced increase in Mt:F across this interval, with the possible exception of proterotheriid litopterns. Proterotheriids are thought to have inhabited forested environments rather than open ones, which raises questions about the selective forces responsible for limb elongation in ungulates. Conversely, notoungulates, which are traditionally thought to have lived in open habitats, show no strong trend of increasing Mt:F across this interval. Our study suggests that the macroevolutionary trend of limb elongation in ungulate-like mammals is not universal and is highly influenced by the evolutionary affinities of the groups being analyzed.

## Introduction

The broad pattern of global climate during the Cenozoic is well-established [1]. The climate was generally warm and humid through the Paleocene and early Eocene, with an abrupt temperature peak at the Paleocene-Eocene boundary [2]; temperatures declined through the middle and late Eocene and then dropped precipitously at the Eocene-Oligocene transition 34 million years ago [3]. Temperatures fluctuated but were relatively steady during the Oligocene and early Miocene and reached a peak around the early-middle Miocene, ca. 17–15 million years ago [4]. Following this climatic optimum, temperatures gradually declined, and aridity increased.

In North America, changes in vegetational structure and mammal ecology associated with these climatic changes have been studied extensively [e.g., 5–10]. During the latter half of the

**Competing interests:** The authors have declared that no competing interests exist.

Cenozoic, ungulates (hoofed herbivores) show two conspicuous trends: increased tooth crown height (hypsodonty) and pronounced 'cursorial' limb modifications (i.e., long, slender, stable limbs). Both of these trends appear to be adaptive responses to increasing proportions of open (i.e., relatively tree-free) habitats. Hypsodonty maintains the functional lifespan of the dentition in open habitats, which are characterized by high levels of exogenous abrasives (grit) as well as abundant phytolith-rich grasses [11–18]. Limbs that are relatively long distally and bear fewer lateral digits are biomechanically more efficient for moving quadrupedally in open habitats since they are lighter, permit more proximal muscle attachments, and result in longer stride lengths, among other factors [19–25]. Hypsodonty increased in many North American ungulate lineages around the early-middle Miocene transition [14], slightly post-dating the appearance of grass-dominated open habitats [26, 27]. An increase in cursoriality (as measured by distal limb elongation) took place some 10 million years earlier than the increase in hypsodonty, during the Oligocene [28; see also S1 File]. Similar patterns have been documented for the Neogene of Eurasia [29].

In contrast to North America, South America was separated from most other continents by large water barriers for nearly the entire Cenozoic. Consequently, few Northern Hemisphere mammal groups were able to disperse to South America during this interval. In their absence, ancient South America ecosystems were filled with mammals exclusive to (or virtually exclusive to) that continent [30–32]. These endemic groups included several clades of ungulate-like herbivores collectively referred to as South American native ungulates (SANUs), as well as xenarthrans (sloths, armadillos, and anteaters), metatherians (marsupials and relatives), and non-therian mammals, among others. The most diverse, abundant, and long-lived SANUs were notoungulates (order Notoungulata) and litopterns (order Litopterna), both of which survived until the late Pleistocene [33–37].

South American notoungulates evolved hypsodont dentitions much earlier than North American ungulates: by the late Eocene in some species and by the late Oligocene in many others [30, 38–41]. Open-habitat grasses were not major components of South American ecosystems prior to the Miocene [42, 43], but plant phytolith ecomorphology indicates that arid conditions and open habitats were present in southern South America as early as the middle Eocene [44, 45]. Thus, although these Eocene open habitats may have resembled grasslands or open shrublands in their vegetational structure, their taxonomic composition was distinct. Mammal ecomorphological data from the early Oligocene of central Chile also indicate that open habitats appeared in South America some 10 million years earlier than in North America [41, 46].

Despite longstanding interest in the so-called 'precocious' hypsodonty of South American notoungulates [17], no study has systematically assessed the evolution of SANU limbs during the middle to late Cenozoic. Shockey and Flynn [47], in their analysis of postcranial remains of Eocene 'isotemnid' notoungulates, noted that the proportion of notoungulates with a superior astragalar foramen (S1 Fig) decreased between the middle Eocene and the late Oligocene, coincident with the appearance of open habitats in South America. Although the function of this foramen (and its associated canal) is unknown both in extinct and extant mammals, it has been proposed that it could have limited full extension at the ankle joint if it transmitted neurovascular tissue, since it is located at the posterior end of the astragalar trochlea (S1 Fig). If so, its loss would permit greater range of motion at the tibioastragalar joint and could represent an adaptation for longer stride length and faster or more efficient locomotion in more open habitats [47]. However, this foramen has been lost in most species of all extant orders of mammals, independent of locomotor habit or cursorial tendencies; it only remains in a few digitigrade species (e.g., the aardvark, *Orycteropus afer*, and the maned wolf, *Chrysocyon brachyurus*), aquatic species (e.g., the neotropical otter, *Lontra longicaudis* and the crabeater

seal, *Lobodon carcinophagus*), and some bears (ursids) [48–51]. This casts doubt on a tight correlation between the loss of this foramen and limb posture or locomotor performance.

In this study, we quantitatively assess limb evolution in SANUs during the late Oligocene to Pleistocene interval using the metatarsal-femur ratio (Mt:F, also known as pes length index), a commonly-used measure of distal limb elongation. Lengthening the distal limb (autopod) relative to the proximal limb (stylopod) is biomechanically advantageous because it increases stride length, which is more efficient [24, 52] and may also enable greater speed [20]. As a result, this variable has been used to study the evolution of cursoriality in a wide variety of mammals [28, 53–55]. Although both ancestry (phylogenetic relationships) and body mass can complicate precise comparisons of Mt:F among groups that are distantly related to one another and/or that differ greatly in size [56], this ratio can be a useful tool for assessing within-group trends and making general comparisons among mammals of roughly similar size [28]. Among SANUs, we focus on notoungulates and litopterns, the most diverse and abundant clades and the only ones with many species of medium to large size (ca. 10–1,000 kg) during this interval [37]. The precise evolutionary relationships of these groups are still uncertain [37], but molecular data from Pleistocene representatives of Litopterna and Notoungulata suggest they are most closely related to Perissodactyla among extant mammals [57–59].

Litoptern relationships are based on McGrath et al. [60, 61], with families Macraucheniidae (Ma) and Proterotheriidae (Pr) indicated. Toxodontian relationships are based on Shockey et al. [62], Bonini et al. [63], and Armella et al. [64]. Typothere relationships are based on Seoane and Cerdeño [65] and Croft and Anaya [66], with *Hemihegetotherium* sp. assumed to be closely related to *Hemihegetotherium achataleptum*. Typothere clades include Hegetotheriidae (He), Hegetotheriinae (H), Interatheriidae (Int), Mesotheriidae (Me), and Pachyrukhinae (P).

## Material and methods

We obtained metatarsal and femur data for 31 genera and 36 species of South American native ungulates pertaining to three major clades: Litopterna, Toxodontia, and Typotheria (S1 Table and Fig 1). Museum specimens were measured to the nearest 0.1 mm using a digital or analog calipers, depending on the size of the specimen. Other measurements were taken directly from the scientific literature or measured from a photo using ImageJ [67]. The sources of each measurement are listed in S1 Table. In most cases (86% of species), measurements were from the same specimen/individual. In the remaining cases, data from different specimens/individuals from the same fossil locality were used to maximize taxon sampling. Femur length was measured parallel to the shaft from the head to the distalmost point, and metatarsal length was measured as greatest length parallel to the long axis of the bone. Each species was assigned a single age based on the specimen that was measured, with ages rounded to the nearest megannum (S1 Table). Typothere notoungulates, toxodont notoungulates, and litopterns were analyzed separately, as were smaller groups in some cases. Statistical analyses and data visualization were conducted using JMP Pro$^{®}$ 14.2 for Mac [68]. Reported slopes are rounded to two significant digits.

## Results

Mean Mt:F is highest for litopterns (0.46; N = 11), followed by typothere notoungulates (0.37; N = 18), and toxodont notoungulates (0.24; N = 7) ($p < 0.05$, Wilcoxon test). None of these three clades shows a pronounced trend in Mt:F across the study interval as indicated by ordinary least-squares regression (Fig 2). Mt:F decreases slightly in toxodonts (slope = -0.0039; $r^2$ = 0.25) and very slightly in typotheres (slope = -0.00088; $r^2$ = 0.013), and it increases slightly in

# Litopterna

- *Cramauchenia insolita*
- *Coniopternium andinus*
- *Llullataruca shockeyi*  (Ma)
- *Theosodon garrettorum*
- *Macrauchenia patachonica*
- *Megadolodus molariformis*
- *Protheosodon coniferus*  (Pr)
- *Anisolophus floweri*
- *Eoauchenia primitiva*
- *Diadiaphorus majusculus*
- *Thoatherium minusculum*

# Notoungulata: Toxodontia

- *Homalodotherium cunninghami*
- *Rhynchippus equinus*
- *Eurygenium pacegnum*
- *Scarrittia canquelensis*
- *Adinotherium ovinum*
- *Nesodon imbricatus*
- *Toxodon* sp.

# Notoungulata: Typotheria

- *Federicoanaya sallaensis*
- *Interatherium extensum*  (Int)
- *Interatherium robustum*
- *Miocochilius anomopodus*
- *Protypotherium attenuatum*
- *Protypotherium australe*
- *Hegetotherium mirabile*  (H)
- *Hemihegetotherium torresi*
- *Hemihegetotherium trilobus*
- *Hemihegetotherium* sp.  (He)
- *Propachyrucos ameghinorum*  (P)
- *Pachyrukhos moyani*
- *Paedotherium* sp.
- *Paedotherium insigne*
- *Trachytherus alloxus*  (Me)
- *Eutypotherium lehmannnitschei*
- *Typotheriopsis internum*
- *Mesotherium cristatum*

**Fig 1. Phylogenetic relationships of SANUs analyzed in this study, grouped by major clade.**

litopterns (slope = 0.0048; $r^2$ = 0.13). As a whole (N = 36), SANUs show a slight upward trend from the late Oligocene to the Pleistocene (slope = 0.0011; $r^2$ = 0.0075).

North American ungulate data (N = 73) are from Janis and Wilhelm [28]. Data for South American native ungulates (N = 36) are provided in S1 and S2 Tables. Regression lines are ordinary least-squared regressions.

Among typotheres, mesotheriids (N = 4) show a slight downward trend in Mt:F (slope = -0.0028; $r^2$ = 0.46), hegetotheriids (N = 8) show a negligible upward trend (slope = 0.00092; $r^2$ = 0.20), as do interatheriids (N = 6; slope = 0.00012; $r^2$ = 0.0001). Among litopterns, macraucheniids (N = 5) show a downward trend (slope = -0.0023; $r^2$ = 0.17), whereas proterotheriids (N = 6) show an upward trend (slope = 0.016; $r^2$ = 0.68).

Only a single SANU, the proterotheriid litoptern *Eoauchenia primitiva*, has Mt:F > 0.65 (Fig 2 and S1 Table), the lower limit among modern ungulates that are generally classified as cursorial (camelids, pecoran ruminants, equids). Most litopterns and some typothere notoungulates have Mt:F between 0.38 and 0.65, below that of extant cursorial ungulates but within the range of extant cursorial (and semiaquatic) carnivorans and rodents. Most notoungulates have Mt:F below 0.38, below that of cursorial carnivorans and rodents.

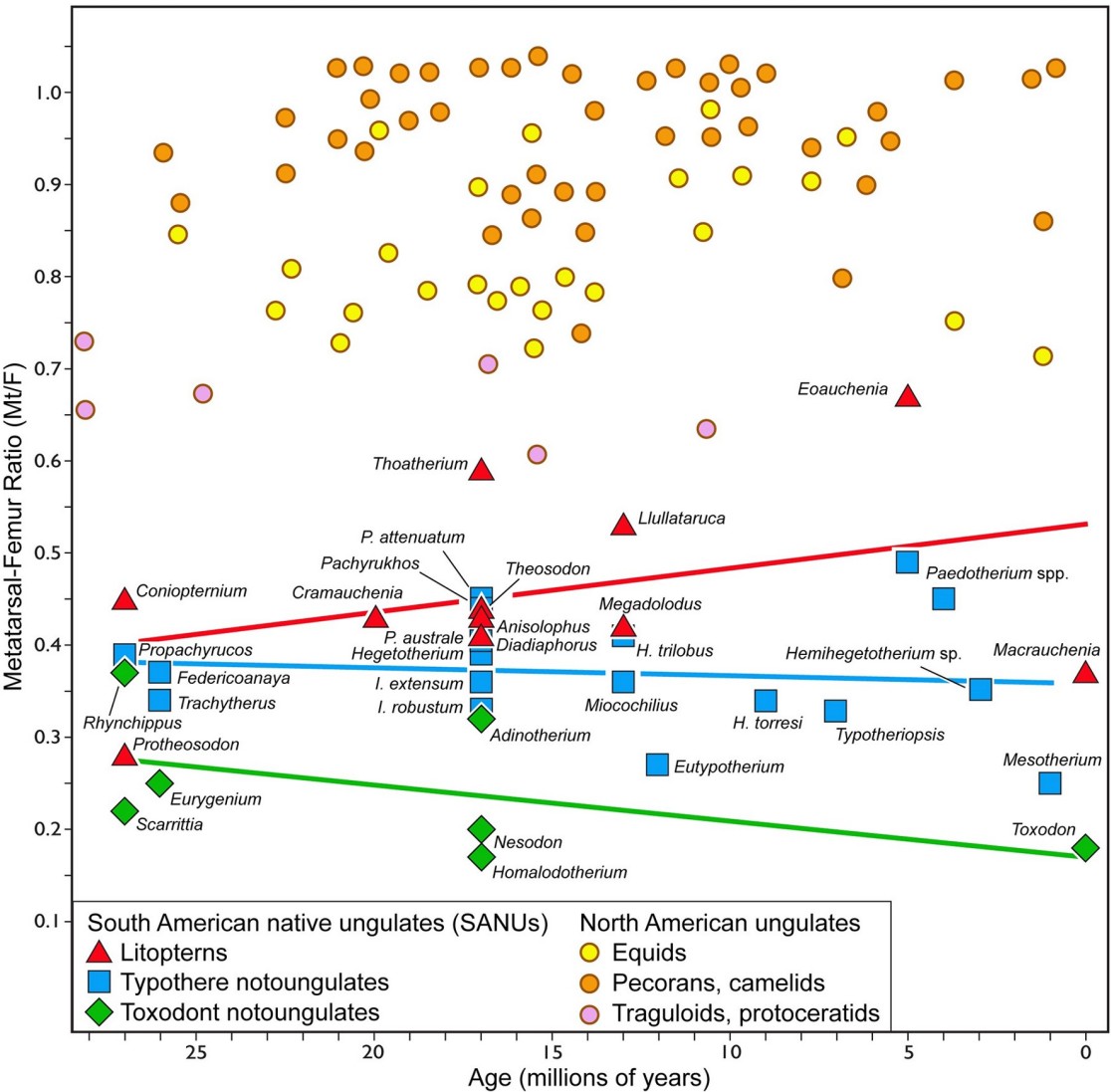

**Fig 2. Metatarsal-femur ratio (Mt:F) in extinct North American ungulates and South American native ungulates.**

## Discussion

As noted previously, North American ungulates show a marked increase in Mt:F around the Oligocene-Miocene transition [28]. A similar pattern has been documented in Eurasia [69, 70]. Our analysis demonstrates that this did not occur in South America, at least from the late Oligocene onwards. Older SANU postcranial remains are rare, but the Mt:F value of 0.33 for the middle Eocene oldfieldthomasiid notoungulate *Allalmeia atalaensis* [71, 72] suggests that no major change in limb proportions occurred among notoungulates during the late Paleogene. (No comparable data are available for litopterns).

It may be that the pattern we observe is the result of biases in our dataset. Although this possibility cannot be excluded entirely, we are aware of no systematic biases in the data that would significantly affect overall patterns. The taxa sampled come from throughout the continent and are relatively evenly distributed among the families sampled; no major families were

unsampled. The taxa studied derive from a variety of habitats, nearly all of which have been reconstructed as at least partially open and suitable for more cursorial species. Our dataset does not sample the majority of extinct notoungulate and litoptern species that have been recorded from this interval, but it appears to be broadly representative.

Another possibility is that ordinal and subordinal patterns are obscuring different trends at the family level that do track climate, habitat, or other variables. This seems to be true to some extent, as the SANU families we analyzed show trends that vary relative to their broader taxonomic group, though such trends seem to reflect ecological specialization and body mass more than habitat per se. It is important to note that most of these more restricted clades are represented by only a few species (generally 4–8; Fig 1 and S1 Table), and interpretations of trends should be considered provisional. Nevertheless, they merit further discussion.

## Family-level patterns

The overall trend in litopterns combines a downward trend among macraucheniids and an upward trend among proterotheriids. The former may relate to body mass (BM), which increases in macraucheniids during the late Cenozoic [61] and has an inverse relationship with Mt:F in our sample; the highest Mt:F pertains to diminutive *Llullataruca shockeyi* [60], while the lowest Mt:F corresponds to one-ton *Macrauchenia patachonica*. An ordinary least-squares regression of Mt:F on log(BM) for macraucheniids has a slope of -0.088 and $r^2$ of 0.707, supporting an inverse relationship between Mt:F and BM in this family. The much weaker relationship between Mt:F and BM in proterotheriids (slope = -0.229; $r^2$ = 0.131) suggests that increasing Mt:F is more closely related to other ecological factors.

Among toxodont notoungulates, the slight decrease in Mt:F through time may also be related to increasing average body mass, since smaller toxodonts ($<$ 100 kg) have higher Mt:F values than larger ones (slope = - 0.088; $r^2$ = 0.538) and the proportion of smaller toxodonts decreases through time (Fig 2 and S3 Table). However, the six species included in our sample pertain to four families (Notohippidae, Leontiniidae, Homalodotheriidae, and Toxodontidae) with varied ecological characteristics, making it unlikely that the trend can be attributed to any single factor.

Among typothere notoungulates, Mt:F decreases through time in mesotheriids. This could reflect increasing body mass [73], though there is only a very weak relationship between Mt:F and BM among the members of the family we analyzed (slope = -0.0300; $r^2$ = 0.025). Specialization for fossoriality [74] is another potential explanation. Hegetotheriids show a very slight upward trend overall, but hegetotheriines show a slight decrease in Mt:F (slope = -0.0028) whereas pachyrukhines show a slight increase (slope = 0.0035). The former trend may relate to increasing body mass, though the relationship between Mt:F and BM is only modest in hegetotheriines (slope = -0.109; $r^2$ = 0.384). The slight increase in pachyrukhines may correlate with cursorial specialization given low body mass variation in this clade (S3 Table).

Interatheres show no trend through time in Mt:F, a surprising result considering that the geologically youngest (middle Miocene) interathere in our analysis, *Miocochilius anomopodus*, has relatively smaller lateral digits than any other notoungulate (Fig 3) and probably further differed in having a subunguligrade stance [75]. Nevertheless, its Mt:F (0.36) is lower than that of some interatheres with less-specialized digits (and presumably a digitigrade stance) such as early Miocene *Protypotherium australe* (0.41; Fig 3). This demonstrates that lateral digit reduction and distal limb elongation were decoupled in notoungulates. To our knowledge, this phenomenon has not been reported in any other group of mammals, though a similar pattern characterized proterotheriid litopterns; the Mt:F of three-toed *Eoauchenia primitiva* from the

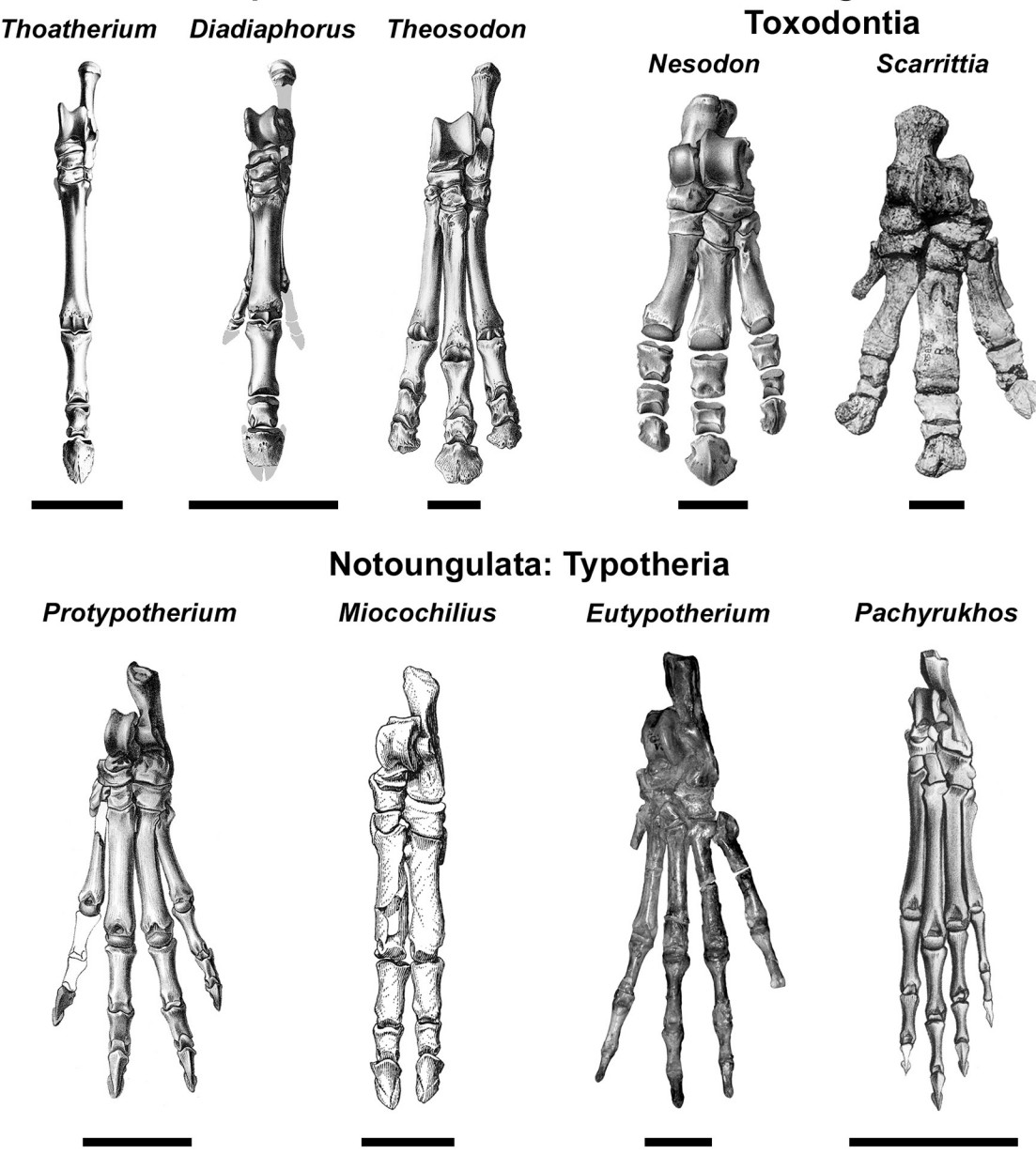

**Fig 3. The pes of representative South American native ungulates.**

Pliocene is slightly greater than that of *Thoatherium minusculum* from the early Miocene, a species with lateral digits proportionally smaller than those of modern horses [76, 77].

Taxa include: *Thoatherium minusculum* (Proterotheriidae), YPPM-VPPU 15719 (reversed; from Scott 1910, pl. XIII, Fig 13 [78]); *Diadiaphorus majusculus* (Proterotheriidae), AMNH 9196 (modified from Scott 1910, pl. V, Fig 2 [78]); *Theosodon lydekkeri* (Macraucheniidae), AMNH 9269 (from Scott 1910, pl. XX, Fig 7 [78]); *Nesodon imbricatus* (Toxodontidae), YPPM-VPPU 15968, (from Scott 1912, pl. XXV, Fig 9 [79]); *Scarrittia canquelensis* (Leontiniidae), AMNH 29585 (from Chaffee 1952, pl. 11, Fig 2 [80]); *Protypotherium australe*

(Interatheriidae), AMNH 9149 (from Sinclair 1909, pl. V, Fig 1 [81]); *Miocochilius anomopodus* (Interatheriidae), UCMP 38091 (from Stirton 1953, pl. 27 [75]); *Eutypotherium lehmannnitchei* (Mesotheriidae), MLP 12–1701 (reversed); *Pachyrukhos moyani* (Hegetotheriidae), AMNH 9481 (reversed; from Sinclair 1909, pl. X, Fig 15 [81]). Scale bars equal 5 cm in upper row (litopterns, toxodonts) and 3 cm in lower row (typotheres).

## Relationship to diet and habitat

Other apparently contradictory evolutionary patterns are revealed when presumed diet and habitat preferences are considered. The SANUs with the most elongate distal limbs, proterotheriid litopterns, were most likely frugivores and/or browsers that lived in forested or partially forested habitats [76, 82–84]. Assuming such dietary interpretations are correct, limb evolution in proterotheriids was not driven by a preference for open habitats, as has been suggested for North American ungulates.

Unlike litopterns, the vast majority of notoungulates in our study are characterized by very hypsodont to hypselodont dentitions [85]. As a result, they are typically reconstructed as open-habitat grazers or mixed feeders [82, 86, 87], though some studies have questioned this interpretation [88–90]. If traditional interpretations of notoungulate paleobiology are correct, it is unclear why they did not develop cursorial specializations to a greater degree, though it may be related to the specific ecological niches they occupied (discussed further below). It is also paradoxical that the sole subungiligrade notoungulate presently known, *Miocochilius anomopodus*, lived in a moist tropical forest [91, 92] rather than an open, arid habitat.

## South American predators

Taken as a whole, there is no evidence that SANUs evolved increasingly cursorial adaptations during the Cenozoic. Although it may be tempting to attribute the lack of such a trend to the unusual predator guild of South America, which was mainly composed of metatherian rather than placental mammals [93–97], Janis and Wilhelm [28] convincingly demonstrated that cursorial predators likely played no role in the early Miocene evolution of cursoriality in North American ungulates. Therefore, the absence of placental predators in South America cannot explain the paucity of cursorial adaptations among its native ungulates. More importantly, pursuit predators actually were present in South America during much of the Cenozoic, but they were flightless birds rather than mammals. These so-called terror birds (phorusrhacids) ranged in height from ca. 60 cm to nearly 3 m [98] and may have been capable of reaching speeds of 65 kph [99], making them large and swift enough to prey on smaller notoungulates and litopterns [100, 101]. If escape from predators were a major factor in the evolution of cursoriality in ungulates, phorusrhacids would have exerted substantial selective pressure on at least small to medium-sized SANUs. The absence of clear cursorial adaptations supports the interpretation that the evolution of mammalian pursuit predators is a consequence rather than a cause of cursoriality in ungulates.

## Litopterns

How can our results be reconciled with current ideas about the paleobiology of SANUs and the evolution of cursoriality? A first step lies in distinguishing between litopterns and notoungulates when analyzing evolutionary patterns and processes. Although both taxa (along with other clades) are referred to as ungulates and lumped together as SANUs, these general terms obscure many fundamental differences between them.

Many litopterns are broadly similar to modern ungulates in limb morphology and, to a large extent, craniodental morphology [32, 37]. Unsurprisingly, they also evolved the most

cursorial adaptations among SANUs. Even though litoptern Mt:F values are generally lower than those of North American ungulates (Fig 2), this may reflect differences in phylogeny and/ or physiology rather than locomotor performance [cf. ref 23]. For example, Janis and Wilhelm [28] noted that the equids in their study tended to have lower Mt:F values than contemporaneous camelid and ruminant artiodactyls and suggested that this could be due to differences in digestive physiology and energy budgets in hindgut-fermenting perissodactyls and foregut-fermenting artiodactyls. The digestive physiologies of litopterns and notoungulates are not known [cf. ref 102], but hindgut fermentation is more likely considering that it is more broadly distributed among mammals [103]. Additionally, molecular data have suggested that SANUs are more closely related to perissodactyls—all of which are hindgut fermenters—than to artiodactyls [57–59]. The relatively low diversity of medium-sized (100s of kg) litopterns and notoungulates compared to small and large ones during the middle to late Cenozoic (S2 and S3 Tables) is similar to that documented for nonruminants in modern East Africa [104] and may constitute circumstantial evidence supporting hindgut fermentation in these groups.

Lower Mt:F values in litopterns could also be related to the prevalence of tridactyly (as opposed to monodactyly) in the group, which itself could be an adaptation for living in closed habitats. Since tridactyly is developed to different degrees in macraucheniids (which have three subequal digits) and proterotheriids (which are functionally monodactyl), it is likely that selective forces differed significantly between the two groups. Nevertheless, there are similarities in other aspects of their limb anatomy. In proterotheriid litopterns, the proximal third phalanx has proportions more similar to those of extinct equids with a "spring foot"—an important adaptation in the evolution of equine unguligrady [105]—than equids that retained three functional digits. The length/width ratio of this element in early Miocene *Thoatherium* and *Diadiaphorus* is 3.05 and 2.95, respectively [78]; these values are much higher than those of the early stem equine anchitheres analyzed by O'Sullivan [values ≤ 2.41; 106], indicating relatively longer, thinner proximal phalanges in these proterotheriids. This is also true for the early Miocene macraucheniid *Theosodon*, whose slightly lower value of 2.58 [78] is still beyond the range of the equids analyzed by O'Sullivan [106]. In contrast to the proximal phalanx, litoptern metatarsals are relatively robust, at least compared to those of artiodactyls. They are classified as broad according to the slenderness index of Morales-García [107], a category otherwise occupied only by the much larger cape buffalo (*Syncerus caffer*) among the 42 extant and extinct artiodactyls they analyzed.

Scott [78] noted that early Miocene proterotheriid litopterns have shorter third metapodials but longer phalanges compared to equids, and Cifelli and Villarroel [108, p. 286] suggested that this might be an alternative strategy for lengthening the distal limb. To test this proposal, they summed the lengths of the tibia, third metatarsal, and proximal phalanx in three proterotheriids (*Thoatherium*, *Diadiaphorus*, and middle Miocene *Megadolodus*) and compared the values to those from 22 extant artiodactyls and perissodactyls. They concluded that all three proterotheriids had very short hind limbs, comparable to extant pigs and peccaries (Suidae and Tayassuidae) as well as tapirs (*Tapirus* sp.). We know of no published data to which the entire proterotheriid hind limb might be compared (i.e., data that include the femur and the entire pes), but Clifford [25] analyzed forelimb proportions in 55 extant and extinct artiodactyls in addition to a limited number of perissodactyls (and carnivorans). Estimations of litoptern stylopod (humerus), zeugopod (radius), and autopod (manus) proportions based on Scott [78] result in values of 31.0%, 27.8%, and 41.2%, respectively, for *Thoatherium* and 31.1%, 30.0%, 38.9% for *Diadiaphorus* (though the former is based on data from two individuals with no elements in common and thus should be considered tentative). These proportions are most similar to those of tayassuids among extant mammals of similar size, congruent with the conclusion of Cifelli and Villarroel [108] that proterotheriids had suoid-like limbs, but more

specific in suggesting greater resemblance to cursorially-adapted tayassuids than suids. Interestingly, Tayassuidae is the same extant group to which Stirton [75] qualitatively compared the limbs of the subunguligrade notoungulate *Miocochilius*, suggesting a parallel "plateau" of limb evolution in litopterns and notoungulates.

The selective forces favoring the evolution of cursoriality in litopterns (especially proterotheriid litopterns) remain obscure, as cursorial adaptations seem to have evolved in closed habitats rather than open ones. This observation apparently contradicts the hypothesis that cursorial modifications in mammals are an adaptive response to open habitats. It may be that such adaptations tend to be favored by natural selection in ungulate-like mammals of medium to large size regardless of habitat in the absence of countervailing selection (e.g., for semifossorial or semiaquatic habits). Another possibility, not mutually exclusive, is that an evolutionary ratchet [45, 109] operates on the postcranium, with cursorial adaptations being highly favored during certain intervals (e.g., dry or seasonal ones that result in a greater proportion of open habitat) or in mixed habitats where browsers (such as litopterns) must traverse open areas between forest patches. Extant white-lipped and collard peccaries (*Dicotyles pecari* and *Pecari tajac*u) are found in a wide variety of both forested and non-forested habitats but are generally restricted to tropical to subtropical regions [110, 111]. In this regard, they may also be reasonable models for proterotheriid litopterns, whose distributions have also been linked to warm climates, though generally ones with humid conditions [112]. The bunodont to bunolophodont dentition of the proterotheriid *Megadolodus* indicates it may have filled an ecological niche very much like that of a modern tayassuid or suid, whereas the lophodont dentitions of other proterotheriids suggest dietary habits closer to those of more folivorous ungulates [32, 108].

## Notoungulates

Most notoungulates, in contrast to litopterns, bear little resemblance to modern ungulates; their postcranial and craniodental morphologies are reminiscent of rodents, rabbits, carnivorans, and even some marsupials [32, 36, 113–116]. This lack of correspondence may explain the absence of cursorial adaptations such as high Mt:F values and fewer functional digits, even though notoungulates may have been living in open habitats. Most notoungulates had an unspecialized ankle joint, consistent with plantigrady or digitigrady (as opposed to unguligrady) and lacked adaptations for locking limb joints like those present in extant ungulates and litopterns [117].

Interatheriid and hegetotheriid typotheres may have been 'functionally cursorial' despite lacking classic cursorial adaptations, as is true of many extant small mammals [54], many of which use a leaping gallop [rather than the horse gallop; 118]. Small size may also explain the scarcity of unguligrady among typotheres, as this feature may be most advantageous in large mammals [56, 119]. Small heteropod footprints (four forefoot digits and three hind foot digits) from the Miocene of La Rioja, Argentina have been considered as probably belonging to a hegetotheriid [120]. The prints are quite similar to those of modern caviids, suggesting a plantigrade to digitigrade gait. Larger hegetotheriids and interatheriids (i.e., ~10 kg; S3 Table) have Mt:F values closer to those of similarly-sized felid and mustelid carnivorans than canids [S1 Table; 28], indicating relatively short distal limbs even compared to other digitigrade and plantigrade mammals. In terms of limb proportions, suoids may also be reasonable postcranial analogs for typotheres of this size [cf. ref 75]. Smaller hegetotheriids have been compared to rodents and interpreted as cursorial with fossorial capabilities [121].

For mesotheriid notoungulates, selective pressures related to a fossorial or semifossorial lifestyle [74, 122] were likely greater than those favoring efficient or rapid locomotion in open

habitats. Larger Neogene typotheres such as mesotheres may have had limb proportions similar to some extinct North American oreodonts (Merycoidodontidae), which were not included in the analysis of Janis and Wilhelm [28] due to their digitigrade stance. Based on published measurements [123], merycoidodontids had Mt:F values of 0.32–0.44, similar to those of many typotheres analyzed here.

The lack of cursorial adaptations in some toxodonts is unsurprising based on other lines of evidence. Homalodotheriids and leontiniids have both been reconstructed as browsers, and neither persisted past the Miocene [62, 80, 124, 125]. Toxodontids had ever-growing dentitions and have been reconstructed as open-habitat grazers [82, 86, 126], though some may have been browsers [90] or have had broad dietary preferences [88]. Toxodontids were tridactyl by the late early Miocene (Fig 3), but their limb proportions did not change appreciably during the remainder of the Cenozoic, at least among larger species (Fig 2). In fact, the geologically youngest species, *Toxodon platensis*, is notably short-legged [127]. If toxodontids occupied ecological niches like large, non-cursorial modern ungulates such as hippos and/or rhinos, it may explain their unusual, robust postcranial morphology and lack of pronounced cursorial adaptations. Testing this hypothesis requires detailed morphofunctional studies of the toxodontid postcranium, which are presently lacking.

## Conclusion

The rich Cenozoic fossil record of the Northern Hemisphere has provided the foundation for much of our understanding about how mammals have responded to changing climates and habitats over the past 66 million years. Nevertheless, such studies are largely based on mammal clades that flourished in the late Cenozoic and dominate modern ecosystems. The fossil record of South America holds great potential for studies of based on different groups of mammals that lack modern representatives or have relatively modest extant diversity. As such, it provides an opportunity to evaluate the degree to which evolutionary hypotheses based on Northern Hemisphere mammals depend on the phylogenetic affinities of the groups being analyzed. This study and others have demonstrated that some South American mammal groups followed very different evolutionary trajectories than their Northern Hemisphere counterparts, resulting in mammal communities that differed in many respects from those of modern South America as well as the Northern Hemisphere, both past and present [85, 91, 97]. Going forward, it is essential to incorporate data from South America and other Southern Hemisphere continents to achieve an integrated understanding of how modern mammal faunas developed and how they might change in the future.

## Supporting information

**S1 Fig. Line drawing of a notoungulate right astragalus illustrating the dorsal astragalar foramen.** The specimen (MLP 75-II-1-9) is from Loma Verde, Argentina, and pertains to a large isotemnid, perhaps *Thomashuxleya*. (A) dorsal view, anterior toward bottom; (B) posterior view, with inferior surface toward top of page. Abbreviations: af: Superior astragalar foramen; at: Astragalar trochlea; fs: Flexor sulcus.
(PDF)

**S1 Table. Femur and third metatarsal (MT3) data for South American native ungulates analyzed in this study.** Species are grouped by family within one of three larger clades: Litopterns (Litop), toxodont notoungulates (N:Tox), and typothere notoungulates (N:Typ). Institutional abbreviations: ACM, Beneski Museum of Natural History, Amherst College, USA; AMNH FM, fossil mammal collection, American Museum of Natural History, New York,

USA; FMNH PM, Fossil Mammals collection, The Field Museum, Chicago, USA; IGM, Instituto de Geociencias y Minería, Bogotá, Colombia; MACN, Museo Argentino de Ciencias Naturales, Buenos Aires, Argentina; MCNAM-PV, vertebrate paleontology collections, Museo de Ciencias Naturales y Antropológicas "J. C. Moyano", Mendoza, Argentina; MLP, Museo de La Plata, Argentina; MNHN-BOL-V, vertebrate paleontology collections, Museo Nacional de Historia Natural, La Paz, Bolivia; UATF-V, vertebrate paleontology collections, Universidad Autónoma "Tomás Frías", Potosí, Bolivia; UCMP, University of California Museum of Paleontology, Berkeley, USA; UF, Florida Museum of Natural History, University of Florida, Gainesville, USA; YPM VPPU, Princeton University Collection, Yale Peabody Museum, New Haven, USA.
(PDF)

**S2 Table. Approximate age of South American native ungulates analyzed in this study.** Ages (to nearest million years, my) are based on the provenance of the specimen(s) from which femur and metatarsal data were collected. The age listed is the value used for calculating regression equations and represents an approximate midpoint of the age for the formation or locality. Species are grouped by family within one of three larger clades: Litopterns (Litop), toxodont notoungulates (N:Tox), and typothere notoungulates (N:Typ).
(PDF)

**S3 Table. Estimated body mass (BM) of South American native ungulates analyzed in this study.** Species are grouped by family within one of three larger clades: Litopterns (Litop), toxodont notoungulates (N:Tox), and typothere notoungulates (N:Typ).
(PDF)

**S1 File. Graph of metatarsal-femur ratio (Mt:F) in Eocene through Pleistocene North American ungulates and large carnivores ($> 7$ kg).** Note the independent increase in Mt:F in several families of ungulates from the Eocene to the early Miocene. From Janis and Wilhelm [17].
(PDF)

## Acknowledgments

We thank Stella Alvarez. Judy Galkin, Alejandro Kramarz, Ruth O'Leary, Marcelo Reguero, Alejo Scarano, and Bill Simpson for facilitating access to museum collections; R. Engelman for compiling data on oreodonts; Christine Janis and an anonymous reviewer for critically reviewing this manuscript and providing constructive criticism and thought-provoking suggestions that improved the final product; and Thierry Smith for handling the manuscript as editor. We also thank Karen Sears and Jon Marcot for the invitation to participate in a symposium on the evolution of the tetrapod limb at the 2017 Annual Meeting of the Society of Vertebrate Paleontology in Calgary, where the initial version of this study was presented.

## Author Contributions

**Conceptualization:** Darin A. Croft, Malena Lorente.

**Formal analysis:** Darin A. Croft.

**Investigation:** Darin A. Croft, Malena Lorente.

**Methodology:** Darin A. Croft, Malena Lorente.

**Writing – original draft:** Darin A. Croft, Malena Lorente.

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
