## [Decision Letter · Decision Letter 0]

29 Jun 2021

PONE-D-21-18278

No evidence for parallel evolution of cursorial limb adaptations among Neogene South American native ungulates (SANUs)

PLOS ONE

Dear Dr. Croft,

Dear Darin,

Two reviewers have accepted to comment on your manuscript. Both of them are very positive. They clearly appreciated the manuscript and regard it as an important contribution. They made several comments and suggestions to help improve it.

In my opinion, most of the issues are minor and do not require much change. The most important issues might be:

1. extending a little the discussion about the diversity of limbs in ungulates and other mammals such as cursorial carnivores and hopper mammals (reviewer 1).

2. the comment about the increase in the Mt:F values from the Oligocene to the Miocene North American ungulates that is not so evident based on Fig. 1 (reviewer 2).

Reviewer 1, Christine Janis, revealed her identity. So, please consider mentioning her name in the acknowledgements if you feel comfortable with this possibility.

We look forward to receiving your revised manuscript.

Kind regards,

Thierry

Thierry Smith, Ph.D.

Academic Editor

PLOS ONE

Journal Requirements:

Reviewers' comments:

Reviewer's Responses to Questions

**Comments to the Author**

1. Is the manuscript technically sound, and do the data support the conclusions?

Reviewer #1: Yes

Reviewer #2: Yes

2. Has the statistical analysis been performed appropriately and rigorously? 

Reviewer #1: N/A

Reviewer #2: Yes

3. Have the authors made all data underlying the findings in their manuscript fully available?

Reviewer #1: Yes

Reviewer #2: Yes

4. Is the manuscript presented in an intelligible fashion and written in standard English?

Reviewer #1: Yes

Reviewer #2: Yes

5. Review Comments to the Author

Reviewer #1: This is a well-written and nicely composed manuscript that furthers our knowledge about mammalian anatomy and adaptations over the course of the Cenozoic. The approach is traditional, and there is no attempt at complex statistical analysis – rather the data are portrayed as a figure where the morphological trends over time in the different lineages can clearly be seen. This approach is entirely appropriate for the type of data obtained and the questions being addressed in this MS, and I do hope that the other reviewer(s) don’t insist on some sort of phylogenetic multivariate analysis which would serve to confabulate, rather than illuminate, the issues here.

The authors address the issue of the commonly used “index of cursoriality” of metatarsal to femur (Mt/F) ratio employed for some modern ungulate groups (primarily equids and ruminant + camelid artiodactyls) and whether it applies to the South American native ungulates (SANUs). These animals certainly appear to be short-legged in comparison with the modern groups, even if they parallel them in some other regards (e.g., loss of digits), and I’m delighted that the authors have actually done this study (for which obtaining data from South American museums was key). They document not only the relatively shorter legs of the SANUs, but also that there is no overall trend for increasing the ratio over time (again in contrast with the modern ungulate lineages, where in North America a ‘step-up’ increase has been documented, correlated with an opening of the habitat in the Oligocene). The conclusions notes that trends noted in modern ungulate lineages should not necessarily be assumed to apply to other groups, and that data from southern continents should be added to our studies in order to better understand evolutionary trends and trajectories.

The figures are clear, and the SI provides comprehensive information about the specimens studied and the original measurements.

I don’t really have any major issues with this MS, which I think sets out what it plans to do very well, and is a good contribution to our knowledge. However, I do think that it could benefit from a little more consideration about the issue of what the Mt/F ratio actually means, and its distribution among extant ungulates.

Basically, a very high ratio (> 0.7) appears to be primarily a feature of equids (at least anchitheres and equines), and ruminant and camelid artiodactyls. I’d like to say that all of these taxa are unguligrade, but -------: secondarily digitigrade camelids don’t count, but most anchitheres have metatarsals as long (or longer than) modern Equus, and they supposedly were sub-unguligrade with a pad foot. Then you have suoids, which are unguligrade but have a low MT/F ratio, so maybe part of this relates to lateral digit reduction (although again here anchitheres upset the apple cart!). (A useful paper to consider here might be Clifford, JVP 2010, although unfortunately it only considers artiodactyls.) But lots of other ungulates (and paenungulates) from diversity of habitats (ceratomorphs, proboscideans, hyraxes) don’t have/never had (as far as we know for hyraxes) long metatarsals. So, a little more discussion about the diversity of today’s ungulates and their limb anatomy could be a good context for why one might expect, or not expect, SANUs to follow the so-called ‘established pattern’. Somewhere here you could also note that more cursorial carnivores also tend to have a higher ratio, even if they never get to be as high as the ungulates.

Another thing to think about a bit more is what a high Mt/F ratio actually means in terms of function and biomechanics. You mention that it relates to the lengthening the distal limb relative to the proximal limb, but perhaps you could be a bit more explicit about relative femur length being relatively constant in ungulates and the “distal limb” usually meaning from the tibia on down. The phalanges are not usually lengthened (but see comment below). Longer limbs do give you longer strides, but in theory lengthening the femur could give you a longer limb (which is what humans and elephants both do). So what is the biomechanics of this anatomy, and why do things that way? This is probably getting into much more than is relevant to this MS, but there’s some stuff that Hildebrand has to say about this that could be added here.

What a longer limb does is give you a longer stride, which isn’t necessarily about speed (see comments below about correlation with habitat, etc.). There may be other reasons to what long limbs or a longer stride, or to *not* want them. If you’re a mediportal mammal, without an elastic energy spring ligament system of recovery in your limbs, it may not benefit you much at all, and then if you’re a hopper you lengthen the tibia more than the metatarsals. All too much for this particular MS, but a little bit of additional discussion could be useful!

I also note that some litopterns, especially Thoatherium, have extremely long proximal phalanges, much more so than unguligrade ungulates (see brief discussion of this in Janis & Bernor, Frontiers in Ecology and Evolution, 2019). Is this somehow compensating for the shorter metapodials? What if one did a calculation of MT+PP/F ratio for all ungulates? (I’m not suggesting that you do that, but maybe mention that they appear to be doing something with their limb morphology quite unlike the equids they’re so often compared to.)

Some small issues:

p. 4: it would be nice to have a brief explanation of why the loss of the superior astragalar foramen was thought to be correlated with “cursoriality”. Also, maybe rephrase a little (especially the final sentence in the paragraph): the correlation of anatomy with habitat is a secondary inference, the primary inference is the correlation of the anatomy with a particular type of locomotor performance (e.g., longer legs enable longer strides, but correlation with open habitat [or even speed] is a secondary inference).

p. 4. Following on from the above, maybe mention that longer strides are actually correlated with locomotor efficiency at all gaits, although they may also enable greater speed. Also, Mt/F ratio declines over a more limited range of sizes than ‘several orders of magnitude’: as can be seen from Table II in Janis & Wilhelm 1993, a fox has a greater Mt/F ratio than a wolf, and a cat a greater one than a lion. (See also Fig 1 in Lovegrove & Mowoe, 2014).

p. 5. Materials and methods. How were these measurements obtained? From the real specimens, by calipers? From the photos, from ImageJ? Also, to the uninitiated it may not be clear here what you mean by ‘endemic ungulate’ as previously you’ve been calling them SANUs.

p. 7. Although Mt/F ratios have only been documented for North American ungulates, note that similar ratios are seen in Old World ruminants and equids.

p. 14. Line 3, should the reference be #35 rather than #6

p. 14. Note that the oreodonts were digitigrade, which was part of the reason that they were not included. Tayassuids and ceratomorphs were also not included, for similar reasons.

p. 14. 7 lines from the bottom ---- the only typo I’ve spotted so far! Should be “broad dietary preferences” (not “board”). Another minor error is in the SI: a couple of times you’ve put “Toxodon sp.” (sp. in italics) rather than “Toxodon sp.” (sp. in Roman type).

Fig. 1. At the expense of overloading the reader with information, perhaps there should be a little more information about who’s who in the North American ungulates (you mention on p. 12 that J&W note lower Mt/F ratios in the equids). I suggest identifying somehow the equids, the protoceratids + traguloids, and then the camelids + pecorans.

S3 Table. You say that body mass estimates are based on femur length. What is the reference group you are using to relate femur length to body mass. Do you have a reference for that?

Reviewer #2: Comments to the author

The spread of grasslands and the evolutionary response of ungulate mammals to the changing environments from closed to open landscapes is well-established in North America. Grasslands spread during the Neogene, and as a response ungulate mammals evolved teeth high higher crowns and longer limbs. However, it is unknown if the same evolutionary trends can be observed in other continents. South America is an excellent study case to evaluate if different evolutionary lineages of herbivorous mammals show similar adaptations to open environments. South America was isolated from other continents during most of the Cenozoic and a large diversity of endemic mammals evolved in relative isolation, including native ungulates (SANUs). This work is the first study to evaluate if SANUs showed the same pattern of limb elongation as North American ungulates in response to the spread of open landscapes. The finding that SANUs do not show a trend of limb elongation in response to the spread of open landscapes is somehow surprising although not completely unexpected, as SANUs have shown different patterns in the evolution of ecomorphological traits in comparison with North American ungulates (for example the early evolution of high-crown teeth in SANUs).

I consider that the manuscript is a valuable contribution to mammal palaeobiology, as it represents the first work to study patterns of limb elongation in South American ungulates across a broad geographical and temporal scales, providing the basis of comparison with evolutionary patterns seen in other continents. Therefore, I consider the work is suitable for publication, although I have several comments that I provide below and should be addressed first.

General comments

The authors discuss how the Mt:F values could be related with body size in some groups of SANUs. I think given these observations it would be appropriate to examine in more detail the possible allometric relationships between the Mt:F and body size.

Does any allometric pattern emerge for Notoungulates and/or Litopterns in a plot of log (Mt:F) vs log (body size)?

Title

I suggest avoiding the use of the acronym SANUs in the title.

Abstract

“None of the groups or subgroups we analyzed show a pronounced increase in Mt:F across this interval”

It would be helpful for the readers to clarify in the abstract if an increase in Mt:F is associated with limb elongation.

Introduction

• “Temperatures fluctuated but were relatively steady during the Oligocene and early Miocene and reached a peak around the early-middle Miocene, 18-16 million years ago”

Please add a reference for the dates of the Miocene Climatic Optimum. Notice that a recent review paper (Steinthorsdottir et al. 2021) on the Miocene climate uses dates of ca. 16.9 – 14.7 Ma for the climatic optimum

Steinthorsdottir, M., et al. (2021). The Miocene: The future of the past. Paleoceanography and Paleoclimatology, 36, e2020PA004037.

• “plant phytolith ecomorphology indicates that arid conditions and open habitats

were present in southern South America as early as the middle Eocene [23, 24].”

I consider is important to clarify (as discussed in the references cited by the authors) that the open habitats in southern South America during the Eocene were shrublands with abundant palms, a flora that was nonanalog to modern open savannas for example. Therefore, even though open habitats might have appeared in southern South America since the Eocene, the flora of these habitats was different from the open habitats that appeared in North America later during the Neogene.

• “Despite longstanding interest in the so-called ‘precocious’ hypsodonty of South American notoungulates”

Here it would be relevant to cite Madden (2015), which in chapters 1 and 2 provides a detailed account on the issue of precocious hypsodonty in South American native herbivores, for the interested reader that may wish to know more about the ‘precocious’ hypsodonty in South America.

Madden (2015). Hypsodonty in Mammals. Evolution, Geomorphology, and the Role of Earth Surface Processes. Cambridge University Press.

Materials and methods

• Please consider adding a figure of a cladogram showing the hypothesis of phylogenetic relationships of the genera of South American native ungulates sampled, showing the subclades that were analysed separately.

• Please clarify if the least square regressions are Ordinary least square (OLS) or generalized least squares (GLS). Given that closely related species might tend to have more similar Mt:F values, the residuals might be correlated and in that case it would be preferable to use GLS over OLS.

Discussion

- “As noted previously, North American ungulates show a marked increase in Mt:F

around the Oligocene-Miocene transition [14].”

Please see the comment below for Fig. 1. Although the trend of increase in Mt:F is reported in the literature, this trend is not clearly visible in the Fig. 1 as shown by the authors. Would a regression (as done for South American taxa) show this trend?

• “The taxa studied derive from a variety of habitats, nearly all of which have been reconstructed as at least partially open and suitable for more cursorial species.”

Please see the comments for Fig. 1, I think the information of habitat (open vs closed) could be indicated by the shape of the points, while the colours correspond to the subclades.

• In addition to the relation of the femur and metatarsal III, the gear ratio of the calcaneus has been also used as a proxy of locomotor habit and posture in carnivorans (e.g. Polly 2010; Polly and Head, 2015). Although the gear ratio has not been studied in detail in ungulates, it could be a relevant metric to explore in future studies. It would be interesting to assess if the gear ratio shows similar patterns than the Mt:F in ungulates from North America and the SANUs.

Polly. 2010. Tiptoeing through the trophics: Geographic variation in carnivoran locomotor ecomorphology in relation to environment. In A. Goswami and A. Friscia (eds.), Carnivoran Evolution: New Views on Phylogeny, Form, and Function

Polly and Head. 2015. MEASURING EARTH-LIFE TRANSITIONS: ECOMETRIC ANALYSIS OF FUNCTIONAL TRAITS FROM LATE CENOZOIC VERTEBRATES. In: Earth-Life Transitions: Paleobiology in the Context of Earth System Evolution. The Paleontological Society Papers, Volume 21,

Figure 1.

The data of Mt:F for North American ungulates showed in Fig. 1 clearly shows that these taxa had a higher Mt:F than South American native ungulates. However, it is not evident that there was an increase in the Mt:F values from the Oligocene to the Miocene. Would a regression line show this trend? (It does not seem it would). Is it necessary to add data from Paleogene North American ungulates to show clearly an increase in Mt:F since the Miocene? Please clarify.

Since the colours of the points and lines differentiate the clades of SANUs (namely litopterns, toxodont notoungulates and typothere notoungulates), the information of whether a taxon has been hypothesized to have lived in a closed or open environment could be indicated by the shape of the points, for example squares and circles).

Figure 2.

Please add a scale to the figure.

Supplementary Information

Table S1

Please explain how the Mt:F value were obtained for the specimens of Scarritia canquelensis that have no measurements of femur and Mt3 and were “computed from indices” as indicated in the comments.

6. PLOS authors have the option to publish the peer review history of their article (what does this mean?). If published, this will include your full peer review and any attached files.

Reviewer #1: **Yes: **Christine M. Janis

Reviewer #2: No

---

## [Author Response · Author response to Decision Letter 0]

1 Aug 2021

Please see our responses in the cover letter and response to reviewers document.

---

## [Editor Report · Decision Letter 1]

5 Aug 2021

No evidence for parallel evolution of cursorial limb adaptations among Neogene South American native ungulates (SANUs)

PONE-D-21-18278R1

Dear Dr. Croft,

Dear Darin,

After carefully looking at the changes you made in the revised manuscript based on the reviewer's comments, I can say that the manuscript has been significantly improved. The introduction is especially informative and pleasant to read. The discussion on the limb proportions and evolution in litopterns has been well developed.  Therefore, I am pleased to inform you that your manuscript has been judged scientifically suitable for publication and will be formally accepted for publication once it meets all outstanding technical requirements.

Kind regards,

Thierry

Thierry Smith, Ph.D.

Academic Editor

PLOS ONE

---

## [Editor Report · Acceptance letter]

9 Aug 2021

PONE-D-21-18278R1 

No evidence for parallel evolution of cursorial limb adaptations among Neogene South American native ungulates (SANUs) 

Dear Dr. Croft:

I'm pleased to inform you that your manuscript has been deemed suitable for publication in PLOS ONE. Congratulations! Your manuscript is now with our production department. 

Kind regards, 

on behalf of

Dr. Thierry Smith 

Academic Editor

PLOS ONE